# The Temporal Change in Ionised Calcium, Parathyroid Hormone and Bone Metabolism Following Ingestion of a Plant-Sourced Marine Mineral + Protein Isolate in Healthy Young Adults

**DOI:** 10.3390/nu16183110

**Published:** 2024-09-14

**Authors:** Marta Kozior, Olusoji Aboyeji Demehin, Michelle Mary Ryan, Shane O’Connell, Philip Michael Jakeman

**Affiliations:** 1Food, Diet and Nutrition, Health Research Institute, University of Limerick, V94 T9PX Limerick, Ireland; martakozior@yahoo.com; 2Marigot Research Centre, Sycamore Court, V92 N6C8 Tralee, Ireland; o.demehin@marigot.ie (O.A.D.); michelle.ryan@marigot.ie (M.M.R.); shane.oconnell@marigot.ie (S.O.); 3Shannon Applied Technology Centre, Munster Technological University, Clash, V92 CX88 Tralee, Ireland

**Keywords:** sustainable diet, plant mineral, plant protein, bone health, post-prandial, young adults

## Abstract

**Background:** An increase in plant-sourced (PS) nutrient intake is promoted in support of a sustainable diet. PS dietary minerals and proteins have bioactive properties that can affect bone health and the risk of fracture. **Methods:** In a group randomised, cross-over design, this study evaluated the post-ingestion temporal pattern of change in arterialised ionised calcium (iCa), parathyroid hormone (PTH), *C*-terminal crosslinked telopeptide of type I collagen (CTX) and procollagen type 1 amino-terminal propeptide (P1NP) for 4 h following ingestion of a novel supplement (SUPP) containing a PS marine multi-mineral + PS protein isolate. A diurnally matched intake of mineral water was used as a control (CON). **Results:** Compared to baseline, the change in iCa concentration was 0.022 (95% CI, 0.006 to 0.038, *p* = 0.011) mmol/l greater in SUPP than CON, resulting in a −4.214 (95% CI, −8.244 to −0.183, *p* = 0.042) pg/mL mean reduction in PTH, a −0.64 (95% CI, −0.199 to −0.008, *p* = 0.029) ng/mL decrease in the biomarker of bone resorption, CTX, and no change in the biomarker of bone formation, P1NP. **Conclusions:** When used as a dietary supplement, or incorporated into a food matrix, the promotion of PS marine multi-mineral and PS protein isolates may contribute to a more sustainable diet and overall bone health.

## 1. Introduction

An increase in plant-sourced (PS) and decrease in animal-sourced (AS) nutrient intake is promoted in support of a sustainable diet [1] and to maintain human and planetary health [2]. However, PS dietary patterns as practised by vegetarians and vegans are associated with a lower bone mineral density (BMD) and increased fracture risk [3]. Therefore, a greater understanding of the functional properties of PS nutrient supplements is required to inform and support the transition to a higher PS nutrient intake for bone health.

Protein and calcium are major components of bone tissue. Protein provides the structural matrix of bone, occupying ~50% of bone volume and one-third of bone mass, and calcium is the dominant mineral within that matrix. Collagen and non-collagenous proteins form the organic matrix of bone, so an adequate dietary protein and calcium intake is essential for optimal acquisition and maintenance of adult bone mass [4]. However, at present, animal-sourced (AS) protein and dairy-based calcium predominate in the Western diet [1,5,6]. 

Bone remodelling describes the normally balanced rate of bone resorption and bone formation that ensures 5–10% of the skeleton per annum is remodelled for essential maintenance and repair. The fracture risk is increased by high remodelling rates or an imbalance (uncoupling) between rates of formation and resorption [7]. Importantly, in the context of PS nutrient intake, a sufficient intake of minerals and protein is required in support of bone remodelling in the adult skeleton [8]. In this respect, legumes are an example of a low-cost PS food of high nutritional value and an excellent source of good-quality protein (20–45% protein), providing high amounts of the essential amino acids such as lysine and leucine [9]. Natural plant sources of calcium are rare, but the multi-mineral extract from the cytoskeleton of the red seaweed *Lithothamnion* sp. (containing 32.6% calcium) is considered an effective PS of mineral for bone health [10]. 

In addition to the supply of mineral and protein in support of bone remodelling, the ingestion of protein and calcium exert an active, temporal change in bone metabolism that favours a post-prandial reduction in the rate of bone resorption and potential for longer-term change in the rate of bone formation [4]. These nutrient bioactivities are related to calcium bioavailability and the subsequent increase in serum ionised calcium [11] and increase in enteroendocrine peptides known to modulate osteoclast function [12] via the entero-osseous axis [13]. Of interest to the present study is the augmentation of post-prandial secretion of enteroendocrine peptides by co-ingestion of calcium [14] and the development of a PS calcium + protein supplement of benefit to bone health.

The principal nutrient modulators of post-prandial bone metabolism under consideration in this report are a plant-sourced marine multi-mineral derived from the cytoskeleton of the red seaweed *Lithothamnion* sp. containing 32.6% calcium (Aquamin F, Marigot Ltd., Cork, Ireland) co-ingested with a plant-sourced legume protein isolate (*Vicia faba* L., Marigot Ltd., Cork, Ireland). We hypothesised that a meal-sized bolus (0.33 g·kg^−1^) of a plant-sourced protein isolate co-ingested with 800 mg of calcium derived from a plant-sourced marine multi-mineral would stimulate a greater post-ingestion calcitropic response compared to a non-bioactive control, leading to a positive effect on bone metabolism. To this end, a study of the acute (0–4 h) temporal change in ionised calcium (iCa), parathyroid hormone (PTH) and the International Osteoporosis Foundation’s (IOF) recommended biomarkers of bone resorption, *C*-terminal crosslinked telopeptide of type I collagen (CTX), and formation, procollagen type 1 amino-terminal propeptide (P1NP), following co-ingestion of a PS marine multi-mineral + legume protein isolate was undertaken in healthy young adult men and women. 

## 2. Materials and Methods

### 2.1. Ethical Approval and Participant Recruitment

This study was granted ethical approval by the University of Limerick Education and Health Sciences Research Ethics Committee (2022_03_06_EHS), and it was conducted in accordance with the ethical standards outlined in the most recent version of the Declaration of Helsinki and registered with clinicaltrials.gov identifier NCT5533502. Potential participants were informed of the risks and benefits before providing written informed consent. Eligibility criteria: (i) aged 18 to 35 years; (ii) healthy (i.e., no current injury, illness, medication, history of chronic disease, known allergies and intolerances; normotensive; non-obese; with normal blood chemistry). All participant volunteers were assessed for eligibility and screened using a medical questionnaire, physical examination, dietary intake record, anthropometry, and body composition analysis (Tanita MC, 180-MA, Tanita Europe B.V., Amsterdam, The Netherlands). Appendix A provides a CONSORT flow diagram of participant enrolment, allocation, follow-up and analysis. Ten participants were recruited to this study, five men and five women. Participant characteristics are presented in Table 1.

### 2.2. Study Design

This study was open to young adult men and women, independent of sex, who satisfied the inclusion criteria. This study had a within-subject, single-blind, block-randomised, repeated-measures, cross-over design comprising two study arms, control (CON) and supplement (SUPP), i.e., subjects acting as their own control. Participant volunteers were randomised at entry to this study. Participants maintained their habitual diet and refrained from purposeful exercise and alcohol consumption for the previous 24 h on two separate occasions at least three days apart. On trial days, participants arrived at the laboratory at 8:00 a.m. following an overnight fast and remained seated throughout the trial. A cannula was inserted into a superficial vein on the dorsal surface of the hand and the hand placed in a heated hand box (air temperature 55 °C) for 15 min to arterialise venous drainage. Following baseline blood draw, participants ingested, in a randomised order, either CON or SUPP within 5 min. Serial samples of arterialised venous blood were then drawn 15, 30, 45, 60, 90, 120, 180 and 240 min post-ingestion.

### 2.3. Supplement Composition

The supplement (SUPP) was composed of a plant-sourced marine multi-mineral (Aquamin F, Marigot Ltd., Cork, Ireland) + plant-sourced protein isolate (*Vicia faba* L., Marigot Ltd., Cork, Ireland). The final composition of the mineral + protein ingested by participants contained 800 mg of calcium (equivalent to 2.45 g of Aquamin F) and 0.33 g of *Vicia faba* L. protein per kilogram of body mass (Table 2) dissolved in 500 mL of water. CON provided an equal volume of mineral water (<15 mg/dL Ca). A detailed analysis of the supplement constituents is provided in Appendix A.

### 2.4. Blood Sample Collection and Analysis

A portion of the whole-blood sample was analysed immediately after collection for iCa and K concentrations by ion-selective potentiometry (I-STAT^®^; Abbott Laboratories, Dublin, Ireland). Intra-assay CV using control solutions was <1% for both iCa and K. Based on baseline samples, the inter-assay CV was <1.5% for iCa and <11% for K. The remaining blood was separated by centrifugation at 10,000× *g* at 4 °C for 5 min and frozen at −80 °C until analysis. In total, 12 of the 180 blood samples were haemolysed and not analysed.

PTH, CTX and P1NP was analysed by 2-site immunometric assay using electrochemiluminescent detection (Roche Cobas e411, Roche Diagnostics, Burgess Hill, UK). The inter-assay CV was <10% for PTH, 5.3% for CTX and 4.5% for P1NP. Active plasma GLP-1_(7–36)_ and total GIP _(1–42)_ were measured using the MSD^®^ metabolic assay (Meso Scale Discovery, Rockville, MD, USA) based on sandwich ELISA and according to the manufacturer’s instructions. The inter-assay CVs were 7.3 and 16.1% for active GLP-1 and total GIP, respectively. 

### 2.5. Treatment of Data and Statistical Analyses

Prior to statistical analysis, potential outliers were assessed by a boxplot, and data were assessed for a normal distribution by the Shapiro–Wilk test (*p* > 0.05), homogeneity of variance by Levene’s test (*p* > 0.05) and Mauchly’s test of sphericity (*p* > 0.05). Data are reported as the mean (SD) unless stated otherwise. An a priori hypothesis for statistical analysis (H_0_) assumed the mean response for SUPP was equal to CON (i.e., µ_SUPP_ = µ_CON_). Temporal change in serially sampled data was analysed by repeated-measures ANOVA, with the level of significance for post hoc tests subject to Bonferroni correction. The overall change in analyte with respect to baseline over the 4 h period post-ingestion was computed by trapezoidal integration and presented as the area under the curve (AUC_0–240_). The difference between treatment AUC_0–240_ was analysed by a paired *t*-test. Cohen’s effect size was calculated using the standardised formula *t*-score/√n. Statistical significance was set at *p* ≤ 0.05. SPSS Version 28 (IBM Corporation, Armonk, NY, USA) was employed for all statistical analyses.

## 3. Results

### 3.1. Temporal Pattern of Change in Ionised Calcium (iCa) Following Ingestion

No change in iCa was observed post-ingestion in CON. iCa increased after 60 min in SUPP with peak values attained at 120 min post-ingestion. These data showed a statistically significant two-way interaction between treatment and time, *F*(8, 72) = 12.35, *p* < 0.001, η_p_^2^ = 0.58. The mean increase in iCa following ingestion was greater in SUPP than CON at the times of 60, 90, 120, 180 and 240 min (Figure 1A,B). When assessed by the integrated AUC_0–240_, SUPP increased the overall ∆iCa response > 5-fold compared to CON (mean difference 7.4 (CI_95%_, 3.16 to 11.7) mmol·min/L, *t*(9) = 3.94, *p* = 0.002).

### 3.2. Temporal Pattern of Change in Parathyroid Hormone (PTH) Following Ingestion

An inverse PTH response to the change in iCa was observed. These data showed a statistically significant two-way interaction between treatment and time, *F*(8, 72) = 5.47, *p* < 0.006, η_p_^2^ = 0.38. The mean decrease in PTH following ingestion was greater in SUPP than CON at the times of 30, 45, 60, 90, 120, 180 and 240 min (Figure 2A,B). When assessed by the integrated AUC_0–240_ ∆PTH, the overall ∆PTH response to SUPP was 66% lower than CON (mean difference −1246 (95% CI, −2365 to −126) pg·min/mL, *t*(9) = −2.517, *p* = 0.016).

### 3.3. Temporal Change in C-Terminal Peptide of Type I Collagen (CTX) and Procollagen Type 1 Amino-Terminal Propeptide (P1NP) Following Ingestion

CTX followed the normal, mid-morning diurnal decrease in CON. A statistically significant two-way interaction between treatment and time, *F*(8, 72) = 6.19, *p* < 0.013, η_p_^2^ = 0.41, was observed post-ingestion. The mean decrease in CTX following ingestion was greater in SUPP than CON at the times of 120, 180 and 240 min (Figure 3A,B). When assessed by the integrated AUC_0–240_ ∆CTX, SUPP decreased the overall ∆CTX response by 70% compared to CON (mean difference −22 (95% CI, −39.7 to −3.30) ng·min/mL, *t*(9) = −2.673, *p* = 0.013.

There was no change in values of procollagen type 1 amino-terminal propeptide (P1NP) at baseline or with respect to the time post-ingestion for CON or SUPP. Baseline values averaged 67.6 (18.7) and 68.1 (2.4) pM for CON and SUPP, respectively.

### 3.4. Temporal Change in Glucose-Dependent Insulinotropic Peptide (GIP) and Glucagon-like Peptide-1 (GLP-1) Following Ingestion

A statistically significant effect of time was observed post-ingestion for GIP (*F*(8, 72) = 4.78, *p* < 0.001, η_p_^2 =^ 0.407) and GLP-1 (*F*(8, 72) = 5.22, *p* < 0.001, η_p_^2^ = 0.43). The mean GIP increased 3-fold from baseline, from 6.4 (2.6) pmol/L to a peak concentration of 17.2 (9.1) pmol/L 30 min post-ingestion (Figure 4A,B). Similarly, the mean GLP-1 increased 4-fold from 0.22 (0.12) pg/mL to a peak concentration of 0.87 (0.28) pg/mL 30 min post-ingestion (Figure 4A,B). 

## 4. Discussion

The interaction between dietary calcium and protein intake and its impact on bone health has been reviewed extensively, with reports of a positive association between increased dietary protein (AS or PS) intake and BMD in healthy men and women and a reduction in the rate of bone loss by supplemental calcium intake [15,16]. Novel PS calcium and protein supplements that positively influence Ca absorption and Ca bioavailability and are beneficial to bone health are desirable, particularly in those who fail to achieve the dietary recommended level of Ca and/or protein [11,15,16,17,18]. Prompted by the global challenge faced in attaining a sustainable diet and to inform and support an increase in plant-sourced (PS) and decrease in animal-sourced (AS) nutrient intake, this study characterised the bone-related nutrient bioactivity following co-ingestion of a novel PS mineral and protein supplement.

The homeostatic control of Ca absorption, excretion, secretion and storage in bone is governed by the requirement to maintain a plasma concentration of ionised Ca (iCa) within a range of 1.1–1.3 mmol/L. This is achieved by the interaction of the calcitropic parathyroid hormone (PTH), 1,25 dihydroxycholecalciferol (1,25 (OH)_2_D_3_) and calcitonin. Counter-regulatory responses, i.e., negative feedback between PTH and 1,25 (OH)_2_D_3_ when iCa decreases, and calcitonin when iCa increases, maintain a tight regulatory control of calcium in the circulation. 

A pilot study comparing the pharmacokinetic and pharmacodynamic responses to the ingestion of an equivalent (720 mg elemental calcium) of plant-sourced marine multi-mineral derived from the cytoskeleton of the red seaweed *Lithothamnion* sp. (Aquamin F) and non-PS (calcium carbonate) reported no difference in ionised or total serum calcium, but a greater lowering of PTH post-ingestion and the calciuric response for PS vs. non-PS calcium in young (mean age 28.8 years), health women [10]. In the present study, the ingestion of a matrix of PS protein and calcium (SUPP) led to an early increase in iCa after 30 min, a rapid rise to a mean peak concentration of 1.30 (Δ 0.053) mmol/L after 120 min post-ingestion, and a 5.3-fold overall 4 h (AUC_0–240_) mean increase in circulating iCa of 8.79 mmol·min/L compared to CON. The counterregulatory effect of an increase in iCa was a suppression of PTH (and presumably renal production of 1,25(OH)_2_D_3_) to reduce the stimulus for osteoclastic bone resorption. Indeed, our observations confirm a temporal pattern of decrease in PTH that mirrored inversely the increase in iCa, resulting in a mean decrease of 66% (−3120 pg·min/mL) in the overall 4 h (AUC_0–240_) circulating PTH compared to CON. 

The acute effect of PS Ca + protein supplementation on bone turnover was measured by the temporal change in validated biomarkers of bone resorption, *C*-terminal peptide of type I collagen (CTX), and formation, procollagen type 1 amino-terminal propeptide (P1NP). Defined by circadian variation in bone turnover markers, bone remodelling exhibits a unimodal diurnal rhythm with a nocturnal peak and daytime nadir [19,20,21]. The diurnal amplitude of bone resorption (CTX) is greater than bone formation and considered the prominent and sensitive biomarker of an acute change in bone remodelling. CTX in the circulation is derived from osteoclastic degradation of type 1 collagen and is widely employed to assess osteoclastic bone resorption and the dynamic change in bone remodelling, e.g., sensitivity to modulation by feeding and specific nutrient intake [22]. In this study, the diurnally matched acute effect of SUPP invoked a 68% decrease in the overall post-prandial AUC_0–240_ for CTX and a statistically significant temporal reduction in CTX from 120 to 240 min. Though not measured, the trend in the observed temporal response suggests that the reduction in bone resorption would extend beyond 240 min. Although SUPP ingestion evoked a transient inhibition of bone resorption, there was no evidence of a post-prandial change in the biomarker of bone collagen formation, P1NP, suggesting a transient shift toward skeletal deposition of calcium in the post-prandial period

The reduction in the appearance of CTX in the circulation probably reflects the calcium-induced reduction in circulating calcitropic PTH. However, the response of bone remodelling to food ingestion per se is also linked to the secretion of the enterogastric hormones glucose-dependent insulinotropic peptide (GIP_1–42_) and glucagon-like peptide-1 (GLP-1_7–36_). Following intake of a standardised mixed meal (498 kcal) or granola bar (260 kcal), levels of GIP and GLP-1 were found to be positively associated with a decrease in CTX [23] and to constitute a link between food intake and bone homeostasis acting via an entero-osseous axis to decrease osteoclast activity [24]. These actions are supported by human studies of an intravenous infusion of GIP that report a ~50% reduction in CTX at 90 min [25]. To affirm the magnitude of enteroendocrine response to co-ingestion of a PS protein and mineral supplement that provided 117 (75) kcal total energy, we observed a temporal increase in GIP and GLP-1 rising ~3 and 4-fold, respectively, from baseline to a peak at 60 min post-ingestion. In the development of a nutrient supplement for bone health, co-ingestion of nutrients that act favourably on bone metabolism via a separate, but complimentary, mechanism of action is an attractive option. Though the present study design could not determine whether co-ingestion of 800 mg PS calcium augmented the post-prandial rise in GIP and GLP-1 evoked by ingestion of 0.33 g/kg body mass of PS protein, or vice versa, a previous study found that co-ingestion of calcium (~1240 mg) with a standardised mixed meal (~300 kcal) resulted in a post-prandial increase of 47% in GIP and a 22% increase in GLP-1 [26]. 

To be considered a good source of Ca, a food or dietary supplement must have a high Ca concentration of high bioavailability, i.e., the proportion of a nutrient present in a food absorbed and utilised. Absorption of the available calcium from foods ranges from <10% to >50% with AS foods, principally dairy, considered to be of high Ca density and bioavailability. In comparison, the Ca density in PS foods is generally lower, and the Ca bioavailability of PS foods or food matrices may be modulated by the presence of anti-nutritional factors (ANFs) such as oxalates and phytates that bind Ca into insoluble and unabsorbable salts by the small intestine, resulting in faecal excretion [27]. However, processing of foods may remove or reduce the presence of ANFs. To this end, the process of isolating the protein from *Vicia faba* L. removed all detectable ANFs present in the native legume. Although the present study primarily targeted PS calcium and protein, the complex relationships among these and other nutrients affecting bone health should be considered. Minerals other than calcium (e.g., magnesium, phosphorous, zinc) are required to optimise bone health. Indeed, in addition to calcium, 2.45 g of SUPP provided ~25% of the DRI for magnesium, <2% of the DRI for phosphorous and <0.1% of the DRI for zinc for adult men and women. Following ingestion, these minerals compete for absorption in the small intestine such that excess consumption of one can impair the absorption of the others and thereby affect the balance of mineral bioavailability. For example, the optimal calcium-to-magnesium ratio from food is 2:1, but it is 12:1 in SUPP (Table 2), raising the possibility of an acute effect on intestinal absorption that could influence the bioavailability of minerals, such as magnesium, contained in SUPP. Calcium and zinc also compete for common absorption sites in the small intestine, though there is limited evidence that supplements containing high levels (~10-fold DRI) of zinc affect calcium absorption when the calcium intake meets the DRI [28], and it is highly unlikely that trace levels of zinc in SUPP would be of concern. Conversely, supplemental calcium (~450 mg/d additional to the DRI) has been shown to reduce net zinc absorption and upset the balance, which may increase the requirement for zinc should a supplemental calcium intake be sustained over prolonged periods [29]. Present only in trace amounts in SUPP, a dietary intake of phosphorous supports the formation and deposition of calcium phosphate within the mineral matrix of bone. Following absorption, phosphorous binds to calcium, reducing the available free calcium within the bloodstream such that an excess phosphorous ingestion is considered hypocalciuric. The complex interplay between minerals highlighted above is further enhanced by the incorporation of minerals with PS protein, or as whole food matrix containing variable amount of minerals, e.g., legumes, nuts and seeds contain high amounts of calcium, while green leafy vegetables, legumes, nuts, seeds and whole grains are rich in magnesium. Furthermore, incorporation into a food matrix results in a slowing of the transit time within the gastrointestinal tract and a change in relative rates of absorption and subsequent bioavailability [11,27,30]. 

## 5. Limitations of the Study

The authors acknowledge that these data pertain to a small sample population of young men and women recruited as healthy volunteers. This study makes no prior assumption as to differences that may be attributed to confounding factors such as each participant’s sex or lifestyle (e.g., diet, physical activity, etc.). As demonstrated by the present study, and that undertaken previously [31], the temporal change in biomarkers of bone metabolism following single-dose ingestion can only provide an indication of the magnitude and time course of the acute post-prandial bioactive effect on bone metabolism and should not be interpreted as a predictor of the long-term outcome on bone health. Qualifying an interventional change in bone health by using the change in bone mineral density may only be possible following a 3–6-month intervention and is only of relevance to the specific population under investigation [32].

## 6. Conclusions

Ingestion of the PS mineral + protein supplement evoked a 70% decrease in the diurnally matched 4 h post-prandial rate of bone resorption and no change in the rate of bone formation. PS nutrient intake is promoted in support of a sustainable diet and to maintain human and planetary health. An adequate dietary protein and calcium intake is essential for the optimal acquisition and maintenance of an adult bone mass. Incorporation of PS mineral and protein into foods as an ingredient, or as a supplement to the diet, is considered beneficial to promote adequate Ca and protein intake and encourage individuals to meet their Ca and protein requirements. The temporal pattern of the response to ingestion of the PS mineral + protein supplement indicates that the mineral extract from the marine red algae, *Lithothamnion* sp., contains a readily bioavailable source of calcium that may be enhanced by the co-ingestion of the *Vicia faba* L. protein isolate, meaning that the Ca bioavailability from foods can be optimised. Though a direct comparison with other sources of calcium + protein intake was not undertaken, the appearance and rise in circulating iCa for the PS mineral + protein supplement occurred earlier in comparison to that of a dairy-based protein and calcium matrix. Incorporating a supplement within the food matrix such as a PS protein isolate should be considered as the matrix effects of PS Ca absorption have a positive influence, meaning that Ca bioavailability is optimised.

## Figures and Tables

**Figure 1 nutrients-16-03110-f001:**
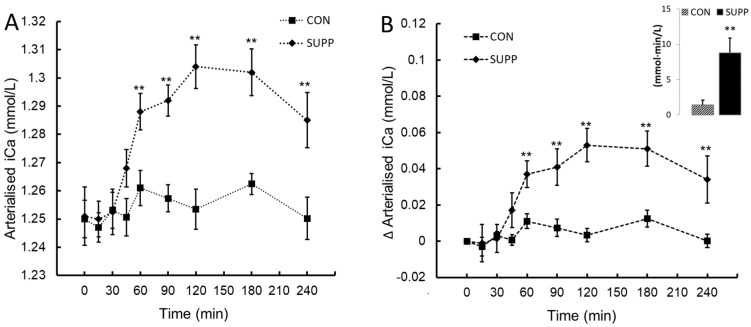
Ionised calcium (iCa; panel (**A**)) and the change in ionised calcium from baseline (∆iCa; panel (**B**)) following ingestion of plant-sourced marine multi-mineral + plant-sourced protein isolate (SUPP) or mineral water (CON). The insert in panel (**B**) reports the integrated area under the curve (AUC_0–240_) of the change from baseline. Data are the mean (SEM). ** Significant difference between SUPP and CON, *p* < 0.05.

**Figure 2 nutrients-16-03110-f002:**
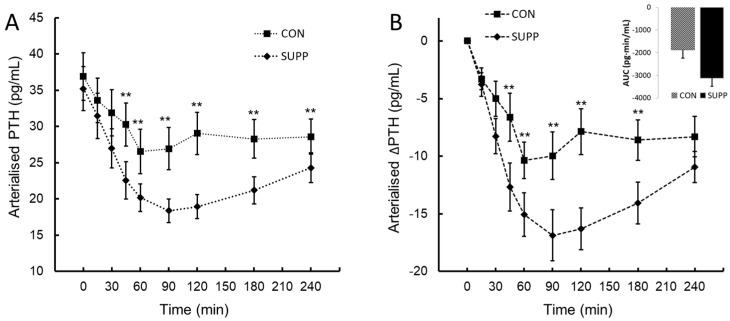
Parathyroid hormone (PTH; panel (**A**)) and the change in parathyroid hormone from baseline (∆PTH; panel (**B**)) following ingestion of plant-sourced marine multi-mineral + plant-sourced protein isolate (SUPP) or mineral water (CON). The insert in panel (**B**) reports the integrated area under the curve (AUC_0–240_) of the change from baseline. Data are the mean (SEM). ** Significant difference between SUPP and CON, *p* < 0.05.

**Figure 3 nutrients-16-03110-f003:**
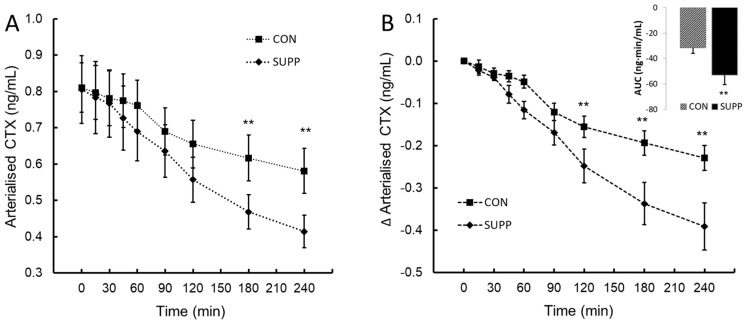
*C*-terminal peptide of type I collagen (CTX; panel (**A**)) and the change in CTX from baseline (∆CTX; panel (**B**)) following ingestion of plant-sourced marine multi-mineral + plant-sourced protein isolate (SUPP) or mineral water (CON). The insert in panel (**B**) reports the integrated area under the curve (AUC_0–240_) of the change from baseline. Data are the mean (SEM). ** Significant difference between SUPP and CON, *p* < 0.05.

**Figure 4 nutrients-16-03110-f004:**
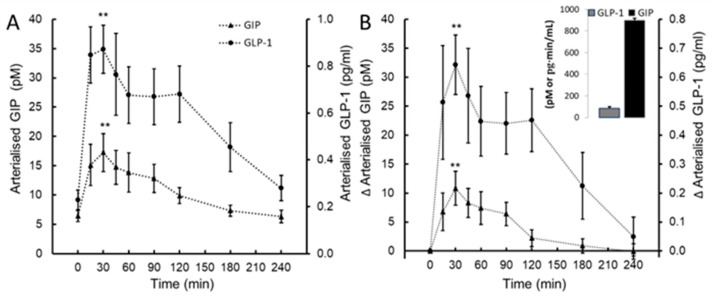
Glucose-dependent insulinotropic peptide (GIP) and glucagon-like peptide-1 (GLP-1; panel (**A**)) and the change in GIP and GLP-1 from baseline (∆GIP, ∆GLP-1; panel (**B**)) following ingestion of plant-sourced marine multi-mineral + plant-sourced protein isolate (SUPP). The insert in panel (**B**) reports the integrated area under the curve (AUC_0–240_) of the change from baseline. Data are the mean (SEM). ** Significant difference from basal, *p* < 0.05.

**Table 1 nutrients-16-03110-t001:** Participant characteristics (*n* = 10; 5 men, 5 women).

	Mean	SD
Age (year)	26.5	4.4
Height (cm)	173.1	9.6
Body mass (kg)	72.5	14.0
Body mass index (kg·m^−2^)	24.1	3.3
Fat mass (%)	20.5	8.2
Fat-free mass (kg)	57.2	9.7
Bone mass (kg)	2.89	0.5

**Table 2 nutrients-16-03110-t002:** Composition of plant-sourced mineral and protein supplement based on participants receiving 800 mg calcium and 0.33 g protein per kilogram of body mass ^1^.

Aquamin F			*Vicia faba* L.		
	g/100 g	mg/2.45 g		Mean	Range
Calcium	32.63	800	g/kg body mass	0.33	
Magnesium	2.59	65	Powder mass (g)	30.9	19.8
Phosphorous	0.044	1.1	Protein mass (g)	23.9	15.3
Zinc	0.2	0.005	Energy (kcal)	117	75

^1^ Note: Full compositional analysis is presented in Appendix A.

## Data Availability

The raw data supporting the conclusions of this article will be made available by the authors on request.

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
