# Peer review of "The Temporal Change in Ionised Calcium, Parathyroid Hormone and Bone Metabolism Following Ingestion of a Plant-Sourced Marine Mineral + Protein Isolate in Healthy Young Adults"

_nutrients, 2024, doi:10.3390/nu16183110_

Round 1

Reviewer 1 Report

Comments and Suggestions for Authors

Dear Authors,

Your study conducted by the authors is interesting. It addresses an important topic concerning the impact of plant-based protein and multi-mineral supplementation on bone health, which is particularly relevant given the increasing popularity of plant-based diets. The study is well-designed, but several significant revisions are necessary.

Major Revisions:

  1. Study Objective: The study’s objective is not clearly defined, and the hypothesis is poorly formulated. The objective of the study should be explicitly stated at the beginning of the "Introduction" section. The hypothesis should be clearly articulated, for example: "We hypothesize that plant-based protein and multi-mineral supplementation will increase ionized calcium levels and decrease parathyroid hormone levels." This should be extended to all analyzed parameters.

  2. Methodology: The methodology section lacks detailed information regarding the randomization process and statistical methods used.

  3. Sample Size: The study includes only 10 participants, which is a very small sample size, limiting the ability to generalize the findings. Such a small sample size increases the risk of Type II errors and decreases the statistical power of the study. This makes it difficult to draw reliable conclusions about the supplement’s effects on a broader population.

  4. Lack of Placebo Control Group: Although there is a control group that received mineral water, the study does not mention a placebo group, which is crucial for minimizing biases and ensuring the validity of the results. A placebo group is essential to distinguish the true effect of the supplement from potential placebo effects.

  5. Use of TANITA for Body Composition Analysis: The use of TANITA for body composition analysis is less accurate compared to more reliable methods such as DXA (Dual-energy X-ray Absorptiometry). TANITA measurements can be significantly influenced by factors such as hydration status, recent food intake, or the menstrual cycle, potentially leading to inaccuracies in body composition analysis.

  6. Absorption and Bioavailability: The study does not address the bioavailability and absorption of the minerals from the supplement, which can vary significantly depending on the participant’s health status, particularly gut health. This is a critical factor influencing the supplement's effectiveness and should be discussed in detail.

  7. Lack of Discussion on Study Limitations: The authors have not included a section discussing the limitations of the study. Acknowledging and discussing limitations such as the small sample size, potential biases, and measurement inaccuracies are essential for maintaining scientific integrity and providing a more balanced interpretation of the results.

  8. Temporal Measurement: The study could benefit from multiple measurements at different time intervals, rather than a single post-ingestion time point. This would help account for variations due to daily fluctuations in metabolism, hydration, and other factors that could influence the results.

Minor Revisions:

  • Data Presentation: The data should be presented more clearly. For instance, tables should include exact values and standard deviations, and graphs should be clear and accompanied by appropriate labels.

Author Response

Response to reviewer # 1 Comments and Suggestions for Authors

The authors thank the reviewer for taking the time to review this M/S

  1. Study Objective: The study’s objective is not clearly defined, and the hypothesis is poorly formulated. The objective of the study should be explicitly stated at the beginning of the "Introduction" section. The hypothesis should be clearly articulated, for example: "We hypothesize that plant-based protein and multi-mineral supplementation will increase ionized calcium levels and decrease parathyroid hormone levels." This should be extended to all analyzed parameters.

Response: The authors welcome these supportive comments and incorporate the following in lines 68 through 77 in the revised M/S.

We hypothesised that a meal-sized bolus (0.33 g·kg-1) of a plant-sourced protein isolate co-ingested with 800mg of calcium derived from a plant-sourced marine multi-mineral would stimulate a greater post-ingestion calcitropic response compared to a non-bioactive control, leading to a positive effect on bone metabolism. To this effect, a study of the acute (0-4h), temporal change in ionised calcium (iCa), parathyroid hormone (PTH) and the International Osteoporosis Foundation’s (IOF) recommended biomarkers of bone resorption, C-terminal crosslinked telopeptide of type I collagen (CTX), and formation, procollagen type 1 amino-terminal propeptide (P1NP), following co-ingestion of a PS marine multi-mineral + legume protein isolate was undertaken in healthy young adult men and women.

This is supported by the statement in section 2.5 of the M/S.’An a priori hypothesis for statistical analysis (H0) assumed the mean response for SUPP was equal to CON (i.e., µSUPP = µCON).’

  1. Methodology: The methodology section lacks detailed information regarding the randomization process and statistical methods used.

Response: Information regarding the randomisation process is provided in section 2.2. Study design. This section is clarified further in the revised M/S as follows.

2.2. Study Design

The study was a within-subject, single-blind, block randomised, repeated measures, cross-over design comprising two study arms control (CON) and supplement (SUPP), i.e., subjects acting as their own control. Participant volunteers were randomised at entry to the study.

Statistical methods are provided in section 2.5. Treatment of data and statistical analysis (extracted below)

2.5. Treatment of data and statistical analyses

Prior to statistical analysis, potential outliers were assessed by boxplot, and data were assessed for normal distribution by Shapiro-Wilk test (P > 0.05), homogeneity of variance by Levene’s test (P > 0.05) and Mauchly’s test of sphericity (P > 0.05). Data are reported as the mean (SD) unless stated otherwise. An a priori hypothesis for statistical analysis (H0) assumed the mean response for SUPP was equal to CON (i.e., µSUPP = µCON). Temporal change in serially sampled data was analysed by repeated measures ANOVA, the level of significance for post-hoc tests subject to Bonferroni correction. The overall change in analyte with respect to baseline over the 4h period post-ingestion was computed by trapezoidal integration and presented as the area under the curve (AUC0-240). Difference between treatment AUC0-240 was analysed by paired t-test. Cohen’s effect size was calculated using the standardised formula (t-score/√n). Statistical significance was set at P ≤ 0.05. SPSS Version 28 (IBM Corporation, Armonk, NY) was employed for all statistical analyses.

The CONSORT FLOW diagram (Figure S1) further supports the above.

  1. Sample Size: The study includes only 10 participants, which is a very small sample size, limiting the ability to generalize the findings. Such a small sample size increases the risk of Type II errors and decreases the statistical power of the study. This makes it difficult to draw reliable conclusions about the supplement’s effects on a broader population.

Response: Agreed. This is common to investigations of this nature. Whilst the P value  provides information on presence or absence of an effect, it will not account for the size of the effect. For comprehensive presentation and interpretation of the study data we have reported  both effect size (ES, ηp2 ) and statistical significance. The statistical power (1-β), where β defines the probability of a Type II error. As you state, the most common reason for this type of error is small sample size, especially when combined with moderately low or low effect sizes (ES). By reporting the ES we provide information on how well the independent variable(s) predict the dependent variable.

  1. Lack of Placebo Control Group: Although there is a control group that received mineral water, the study does not mention a placebo group, which is crucial for minimizing biases and ensuring the validity of the results. A placebo group is essential to distinguish the true effect of the supplement from potential placebo effects.

Response: The study design (articulated and clarified in the revised section 2.2.) does not incorporate 2 separate study groups, rather a within-subject repeated measures cross-over design with participants acting as their own control. The relevant section is extracted for the reviewer’s consideration.

The study was a within-subject, single-blind, block randomised, repeated measures, cross-over design comprising two study arms control (CON) and supplement (SUPP), i.e., subjects acting as their own control.

  1. Use of TANITA for Body Composition Analysis: The use of TANITA for body composition analysis is less accurate compared to more reliable methods such as DXA (Dual-energy X-ray Absorptiometry). TANITA measurements can be significantly influenced by factors such as hydration status, recent food intake, or the menstrual cycle, potentially leading to inaccuracies in body composition analysis.

Response: Body composition analysis was undertaken to provide relevant measures of anthropometric data pertaining to the sample of the populace volunteers. Body compositional data was not considered an independent factor in the analysis. The research group would be aware of the conditions and errors in BIA vs. DXA measurement of body composition. ( DOI: 10.1007/s00421-011-2010-4 )

  1. Absorption and Bioavailability: The study does not address the bioavailability and absorption of the minerals from the supplement, which can vary significantly depending on the participant’s health status, particularly gut health. This is a critical factor influencing the supplement's effectiveness and should be discussed in detail.

Response: The eligibility criteria for participants were stated in section 2.1. extracted below.

Eligibility criteria: (i) aged 18 to 35 y; (ii) healthy (i.e., no current injury, illness, medication, history of chronic disease, known allergies and intolerances, normotensive, non-obese, with normal blood chemistry). All participant volunteers were assessed for eligibility and screened using a medical questionnaire, physical examination, dietary intake record, anthropometry, and body composition analysis (Tanita MC, 180-MA, Tanita United Kingdom Ltd).

The participants were considered ‘healthy’ in accordance with these criteria. The participant’s gut health was not assessed in this study design, so it is not possible to comment or discuss the influence of gut health as requested.

  1. Lack of Discussion on Study Limitations: The authors have not included a section discussing the limitations of the study. Acknowledging and discussing limitations such as the small sample size, potential biases, and measurement inaccuracies are essential for maintaining scientific integrity and providing a more balanced interpretation of the results.

Response: We acknowledge this omission and thank the referee for due diligence. The following has been included (lines 328 through 331) in the revised M/S.

The authors acknowledge that these data pertain to a small sample population of young men and women recruited as healthy volunteers. The study makes no prior assumption as to differences that may be attributed to confounding factors such as the participant’s sex or lifestyle (e.g. diet, physical activity, etc.). 

Accuracy of measurement of the dependent variables, by calibration against reference standards used by the manufacturers, is not normally quoted within the methods section of a M/S but incorporated within the ‘term in accordance with the manufacturer’s instructions/protocols’ (see section 2.4.). Neither is the specificity, detection limits and repeatability of assay from the manufacturer. However, the inter-assay CV of conducting the assay in-house is reported to indicate the reliability/reproducibility of the assay by the researchers ‘in-house’ (see section 2.4.).

  1. Temporal Measurement: The study could benefit from multiple measurements at different time intervals, rather than a single post-ingestion time point. This would help account for variations due to daily fluctuations in metabolism, hydration, and other factors that could influence the results.

Response: With respect. The design is a temporal study (9 timepoints spanning a 4h period) conducted at the same time of day, under the same pre-study conditions in a controlled laboratory environment. Section 2.2. contains this information and is extracted below for the reviewer’s consideration.

Participants maintained their habitual diet and refrained from purposeful exercise and alcohol consumption for the previous 24h on two separate occasions at least three days apart. On trial days, participants arrived at the laboratory @ 8:00am following an overnight fast and remained seated throughout the trial. A cannula was inserted into a superficial vein on the dorsal surface of the hand and the hand placed in a heated hand box (air temperature 55°C) for 15 min to arterialise venous drainage. Following baseline blood draw, participants ingested, in randomised order, either CON or SUPP within 5 minutes. Serial samples of arterialised venous blood were then drawn 15, 30 ,45 ,60 ,90 ,120 ,150, 180- and 240-min post-ingestion.

Minor Revisions:

  • Data Presentation: The data should be presented more clearly. For instance, tables should include exact values and standard deviations, and graphs should be clear and accompanied by appropriate labels.

Response: With respect. The data tables contain exact values. The data in Table 1 are reported with SD because it is appropriate to describe the data distribution of the sample population. The data in Table 2 report the actual values of constituents of Aquamin F corresponding to a constant amount of calcium – i.e., there is no distribution- and the mean and range of powder mass, protein mass and energy provided that corresponds to the range of participant’s body mass.

We contend the Figures are constructed with clarity and accompanied by appropriate labels  legends consistent with the publisher’s guidelines.

Reviewer 2 Report

Comments and Suggestions for Authors

In this study, the authors administered 800mg calcium and 0.33g protein per kilogram of body mass (more and less 800 mg x 70 = 56 grams) in a cohort of volunteers (SUPP) (based on 2 compounds, Aquamin F and Vicia faba L). The control group  (CON) received <15mg x 5 = <75 mg of Ca dissolved in mineral water. They found that compared to baseline, the change in iCa concentration was 0.022 (95%CI, 0.006 to 0.038, p = 0.011) mmol/l greater in SUPP than CON, resulting in a -4.214 (95%CI, -8.244 to -.183, p = 0.042) pg/mL mean reduction in PTH, a -0.64 (95%CI, -0.199 to -0.008, p = .029) ng/mL decrease in the biomarker of bone resorption, CTX, and no change in biomarker of bone formation, P1NP. The authors concluded their study by stating that when used as a dietary supplement, or by incorporation into a food matrix, the promotion of plant-sourced (PS) nutrient marine multi-mineral and PS protein isolates may contribute to a more sustainable diet and overall bone health.

The paper is potentially interesting; however, there are some flaws.

- I did not understand the number of recruited individuals in the SUPP and CON groups. Did the authors recruit 10 SUPP and 10 CON? If yes, the total n is quite low.

-  Were the recruited individuals men and women? If yes, the n is really quite low.

- The sex effect should be considered in the statistics, accordingly.

- A repeated measure ANOVA could be used including F and dF in the results.

- 56 grams of Ca is a huge dose compared to mineral water only. Did the authors consider the use of solid food or liquids containing high amounts of Ca (cheese/milk for example) as a further control?

Comments on the Quality of English Language

 Minor editing of English language required.

Author Response

Response to reviewer # 2 Comments and Suggestions for Authors

The authors thank the reviewer for taking the time to review this M/S

In this study, the authors administered 800mg calcium and 0.33g protein per kilogram of body mass (more and less 800 mg x 70 = 56 grams) in a cohort of volunteers (SUPP) (based on 2 compounds, Aquamin F and Vicia faba L). The control group  (CON) received <15mg x 5 = <75 mg of Ca dissolved in mineral water. They found that compared to baseline, the change in iCa concentration was 0.022 (95%CI, 0.006 to 0.038, p = 0.011) mmol/l greater in SUPP than CON, resulting in a -4.214 (95%CI, -8.244 to -.183, p = 0.042) pg/mL mean reduction in PTH, a -0.64 (95%CI, -0.199 to -0.008, p = .029) ng/mL decrease in the biomarker of bone resorption, CTX, and no change in biomarker of bone formation, P1NP. The authors concluded their study by stating that when used as a dietary supplement, or by incorporation into a food matrix, the promotion of plant-sourced (PS) nutrient marine multi-mineral and PS protein isolates may contribute to a more sustainable diet and overall bone health.

The paper is potentially interesting; however, there are some flaws.

  1. I did not understand the number of recruited individuals in the SUPP and CON groups. Did the authors recruit 10 SUPP and 10 CON? If yes, the total n is quite low.

Response: Agreed. This is common to investigations of this nature. To clarify, this is a within-subject design, i.e., more powerful than an independent 2 x group design and the text in section 2.2. has been revised as per below.

2.2. Study Design

The study was a within-subject, single-blind, block randomised, repeated measures, cross-over design comprising two study arms control (CON) and supplement (SUPP), i.e., subjects acting as their own control.

  1. Were the recruited individuals men and women? If yes, the n is really quite low.
  2. The sex effect should be considered in the statistics, accordingly.

Response: The authors acknowledge that these data pertain to a small sample population of young men and women recruited as healthy volunteers. A within-subject design, the study makes no prior assumption as to differences that may be attributed to confounding factors such as the participant’s sex or lifestyle (e.g. diet, physical activity, etc.). 

  1. A repeated measure ANOVA could be used including F and dF in the results.

Response: ANOVA® was employed. The relevant section of the M/S is extracted below for the reviewer’s consideration.

2.5. Treatment of data and statistical analyses

Prior to statistical analysis, potential outliers were assessed by boxplot, and data were assessed for normal distribution by Shapiro-Wilk test (P > 0.05), homogeneity of variance by Levene’s test (P > 0.05) and Mauchly’s test of sphericity (P > 0.05). Data are reported as the mean (SD) unless stated otherwise. An a priori hypothesis for statistical analysis (H0) assumed the mean response for SUPP was equal to CON (i.e., µSUPP = µCON). Temporal change in serially sampled data was analysed by repeated measures ANOVA, the level of significance for post-hoc tests subject to Bonferroni correction. The overall change in analyte with respect to baseline over the 4h period post-ingestion was computed by trapezoidal integration and presented as the area under the curve (AUC0-240). Difference between treatment AUC0-240 was analysed by paired t-test. Cohen’s effect size was calculated using the standardised formula (t-score/√n). Statistical significance was set at P ≤ 0.05. SPSS Version 28 (IBM Corporation, Armonk, NY) was employed for all statistical analyses.

The results report the F(dF) for all analyses in the following format throughout (extracted from Results, line 149-151)

 These data shown a statistically significant two-way interaction between treatment and time, F(8, 72) = 12.35, P < .001, ηp2 = 0.58.

  1. 56 grams of Ca is a huge dose compared to mineral water only.

Response: Unsure as to the origin of 56g of calcium? The supplement composition and amount ingested by the participants is stated in section 2.3 (extracted below). The amount of calcium is highlighted in BOLD.

‘The final composition of the mineral + protein ingested by participants contained 800mg of calcium (equivalent to 2.45g of Aquamin F) and 0.33g of Vicia faba L protein per kilogram of body mass (Table 2) dissolved in 500ml of water.’

Table 2. 1Composition of plant-sources mineral and protein supplement based on participant receiving 800mg calcium and 0.33g protein per kilogram of body mass.

Aquamin F

Vicia faba L

ppm

g/100g

mg/2.45g

mean

range

Calcium

326300

32.63

800

g/kg body mass

0.33

Phosphorus

110

0.01

0.3

Powder mass (g)

30.9

19.8

Potassium

242

0.02

0.6

Protein mass (g)

23.9

15.3

Sodium

3839

0.38

9.4

Energy (kcal)

117

75

Did the authors consider the use of solid food or liquids containing high amounts of Ca (cheese/milk for example) as a further control?

Response: Thank you for this comment. Matrix effects are one for the future!

Such an option was outside the remit of the current study. However, the discussion refers to one of our previous studies of a dairy-based protein and calcium matrix (lines 289 through 295, [11]).

Reviewer 3 Report

Comments and Suggestions for Authors

The topic of the paper is interesting and might be of interest to the readers of Nutrients journal. The introduction provides an overview of the field and the need for conducting present study. The experimental design of the study is well-described and the obtained results are presented clearly. However, there are some minor issues that should be clarified prior publication:

1. The present title of the paper is too long; I would suggest to use shorter title. 

2. I would recommend to add a figure with schematic presentation of the study design that would improve the readability of the paper. 

Author Response

Response to reviewer # 3 Comments and Suggestions for Authors

The authors thank the reviewer for taking the time to review this M/S

Comments and Suggestions for Authors

The topic of the paper is interesting and might be of interest to the readers of Nutrients journal. The introduction provides an overview of the field and the need for conducting present study. The experimental design of the study is well-described and the obtained results are presented clearly.

The authors thank the reviewer for these supportive comments.

However, there are some minor issues that should be clarified prior publication:

  1. The present title of the paper is too long; I would suggest to use shorter title. 

Response: The authors considered the requirement for the title to convey the nature of the study, the material under investigation, and the population. Though it would be preferable to retain the current title a shorter version, the following would be acceptable.

‘The temporal change in ionised calcium, parathyroid hormone and bone metabolism following ingestion of a plant-sourced marine mineral and protein isolate in healthy young adults.’

  1. I would recommend to add a figure with schematic presentation of the study design that would improve the readability of the paper. 

Response: Section 2.2. Study Design has been modified to clarify. The authors considered the CONSORT flow chart (Supplemental Figure S1) to provide a schematic of the study design. Figure S1 could be moved to the main body of the M/S should further clarity be warranted.

Round 2

Reviewer 1 Report

Comments and Suggestions for Authors

Dear Authors,

Thank you for your valuable work exploring calcium bioavailability from plant-based supplements. Your study makes an important contribution to the field of bone health, but I believe there are areas in your manuscript that could be enhanced in the Discussion and Limitations sections. I understand that the scope of your current study cannot be changed, but addressing the following points in your discussion would underscore the need for more detailed future research in this area:

  1. Expand on Overall Mineral Metabolism: Your study primarily examines calcium bioavailability, but it does not address how supplementation might affect the overall mineral balance, including other critical minerals such as magnesium, phosphorus, and zinc. These minerals are vital for bone health and metabolism, and their interactions with calcium supplementation could have significant implications. Including this in your discussion would highlight the complex nature of mineral metabolism and emphasize the need for further research on the broader impact of supplementation.
  2. Comparison with Animal Proteins: The manuscript focuses on plant-based protein sources, but it lacks a comparison with animal proteins, which might influence calcium bioavailability differently due to variations in amino acid profiles, hormonal interactions, and micronutrient content. Discussing the potential differential effects of animal versus plant proteins on calcium absorption could provide a broader context for your findings and highlight an important area for future study.
  3. Study Population and Broader Applicability: Your study involves young women, which limits the generalizability of the results, particularly to peri- and postmenopausal women who are at increased risk of osteoporosis due to hormonal changes. Addressing this limitation in the Discussion and suggesting research focused on these age groups would provide greater insight into the potential benefits and limitations of the supplement in different populations.
  4. Long-Term Impacts on Bone Health: The study assesses short-term bioavailability, which does not provide information on the long-term effects of the supplement on bone health, including bone remodeling and changes in bone mineral density. Prolonged metabolic changes influence osteoporosis; therefore, a brief acknowledgment of this limitation and a call for long-term studies would add significant value to the paper.
  5. Complex Mechanisms Underlying Osteoporosis: Osteoporosis is a multifactorial disease involving not just calcium deficiency but also hormonal, genetic, dietary, and lifestyle factors. Discussing these complex mechanisms, particularly the influence of sex hormones and other factors like vitamin D levels, would enrich the context of your findings and underscore the multifaceted nature of bone health.
  6. Detailed Limitations of the Study: Although the manuscript mentions the small sample size as a limitation, a dedicated "Limitations of the Study" section could further elaborate on other constraints, such as the metabolic diversity of participants, lack of control over dietary factors, and limitations of the bioavailability measurement methods used. Highlighting these aspects would clarify the study's boundaries and reinforce the necessity for meticulous and comprehensive future research.

Recommendations for Minor Revisions:

  • Expand the discussion to address the impact of supplementation on overall mineral metabolism, including magnesium, phosphorus, and zinc.
  • Address the potential differences in calcium bioavailability between plant and animal proteins and suggest further research in this area.
  • Discuss the limitations of the study population, particularly the focus on young women, and propose future studies involving peri- and postmenopausal women.
  • Highlight the importance of long-term studies to assess the effects of the supplement on bone remodeling and mineral density.
  • Include a broader discussion on the complex mechanisms contributing to osteoporosis, emphasizing hormonal and other non-calcium-related factors.
  • Develop a more comprehensive "Limitations of the Study" section that details sample size constraints, metabolic variability, and methodological limitations.

Author Response

Response to reviewer # 1 – round 2 Comments and Suggestions for Authors

The authors thank the reviewer for taking the time to re-review this M/S

Dear Authors,

Thank you for your valuable work exploring calcium bioavailability from plant-based supplements. Your study makes an important contribution to the field of bone health,

Response: The authors thank you for this positive overall view of the current research.

 but I believe there are areas in your manuscript that could be enhanced in the Discussion and Limitations sections. I understand that the scope of your current study cannot be changed, but addressing the following points in your discussion would underscore the need for more detailed future research in this area:

  1. Expand on Overall Mineral Metabolism: Your study primarily examines calcium bioavailability, but it does not address how supplementation might affect the overall mineral balance, including other critical minerals such as magnesium, phosphorus, and zinc. These minerals are vital for bone health and metabolism, and their interactions with calcium supplementation could have significant implications. Including this in your discussion would highlight the complex nature of mineral metabolism and emphasize the need for further research on the broader impact of supplementation.

A welcome guidance. However, it is probably that discussion of these points cannot be satisfied fully by the objectives and study design of the present investigation that, as stated, is an investigation of the calcitropic response to ingestion of a PS mineral+protein supplement in support of attainment of a sustainable diet. The study was not designed to provide a review of the complexity of mineral metabolism and the differential effect such a response may evoke in specific populations and over the short and long term. That said, an attempt has been made to address this recommendation in the revised m/s, lines 333 through 354, by reference to, and relevance of, the actual mineral content in SUPP, i.e., without overly speculating or extending to mineral complexes/composition(s) residing outside the scope of this study. To bring these specific minerals into focus Table 2 has been modified to highlight the mineral content under discussion.

‘Although the present study primarily targeted PS calcium and protein, the complex relationships among these and other nutrients affecting bone health should be considered. Minerals other than calcium (e.g., magnesium, phosphorous, zinc) are required to optimise bone health. Indeed, in addition to calcium 2.45g of SUPP provided ~25% of the DRI for magnesium, <2% of the DRI for phosphorous, and <0.1% DRI for zinc for adult men and women. Following ingestion, these minerals compete for absorption in the small intestine such that excess consumption of one can impair the absorption of the other and thereby affect the balance of mineral bioavailability. For example, the optimal calcium to magnesium ratio from food is 2:1 but 12:1 in SUPP (Table 2), raising the possibility of an acute effect on intestinal absorption that could influence the bioavailability of minerals, such as magnesium, contained in SUPP. Calcium and zinc also compete for common absorption sites in the small intestine, though there is limited evidence that supplements containing high levels (~10-fold DRI) of zinc affect calcium absorption when calcium intake meets the DRI [28], and highly unlikely that trace levels of zinc in SUPP would be of concern. Conversely, supplemental calcium (~450mg/d additional to the DRI) has been shown to reduce net zinc absorption and balance, which may increase the requirement for zinc should supplemental calcium intake be sustained over prolonged periods [29]. Present only in trace amounts in SUPP, dietary intake of phosphorous supports the formation and deposition of calcium phosphate within the mineral matrix of bone. Following absorption phosphorous binds to calcium reducing the available free calcium within the bloodstream such that an excess phosphorous ingestion is considered hypocalciuric.’

  1. Comparison with Animal Proteins: The manuscript focuses on plant-based protein sources, but it lacks a comparison with animal proteins, which might influence calcium bioavailability differently due to variations in amino acid profiles, hormonal interactions, and micronutrient content. Discussing the potential differential effects of animal versus plant proteins on calcium absorption could provide a broader context for your findings and highlight an important area for future study.

Response: A similar response to above. We (e.g. https://doi.org/10.3390/nu15194211) , and others (e.g. https://doi.org/10.1007/s00726-018-2640-5), have examined the variation in amino acid profiles of AS and PS nutrient intake and have undertaken detailed examination of the post-prandial aminoacidaemia and related circulatory hormonal response to AS and PS proteins, protein extracts and hydrolysates and, to a lesser extent AS calcium. The variability in these data is large and becomes more and more complex as you consider food matrix effects etc., etc. Thus, we would find a full discussion of potential differential effects of AS and PS products as you suggest challenging and outside the scope of the specific study objectives submitted for review. The argument is supported by a recent and well-respected review by Rizzoli and Chevalley [3] provided within the M/S to inform the reader. Indeed, it was a privilege to listen to and discuss with Rizzoli at the nutrition and bone health meeting in San Diago earlier this year.

That said, an attempt has been made to address this recommendation in the revised m/s, lines 323 through 354, by reference to and relevance of the actual mineral content in SUPP

‘To be considered a good source of Ca, a food or dietary supplement must have a high Ca concentration of high bioavailability, i.e., the proportion of a nutrient present in a food absorbed and utilized. Absorption of the available calcium from foods range from <10% to >50% with AS foods, principally dairy, considered to be of high Ca density and bioavailability. By comparison, Ca density in PS foods is generally lower and Ca bioavailability of PS foods or food matrices may be modulated by the presence of anti-nutritional factors (ANF) such as oxalates and phytates that bind Ca into insoluble and unabsorbable salts by the small intestine, resulting in faecal excretion [27]. However, processing of foods may remove or reduce the presence of ANF. To this effect the process of isolating the protein from Vicia faba L removed all detectable ANF present in the native legume.’

  1. Study Population and Broader Applicability:Your study involves young women, which limits the generalizability of the results, particularly to peri- and postmenopausal women who are at increased risk of osteoporosis due to hormonal changes. Addressing this limitation in the Discussion and suggesting research focused on these age groups would provide greater insight into the potential benefits and limitations of the supplement in different populations.

Response: As stated previously, this is a study of young adult women and men that does not engage in age comparison within or between the sexes. The acute and chronic use of protein+mineral supplements on bone health in peri- and postmenopausal women has been addressed thoroughly, reviewed and referred within the cited references [4], [8], and [11] and reported previously by our group [31,32]. The study is limited to a convenience sample of a specific population and not generalisable. That said, the study was designed as a first step to investigate whether PS mineral and protein supplement could have an acute, postprandial, positive effect on bone metabolism, which appears to be the case. As per our previous work, the next stage is to apply these outcomes to a longer term (i.e., 24-week [32]) study in the populations identified at greatest risk. We have therefore crafted an additional section related to these points into the discussion of the limitations to the study, lines 365 through 372.

‘As demonstrated by the present study, and that undertaken previously [31], the temporal change in biomarkers of bone metabolism following single dose ingestion can only provide an indication of the magnitude and time course of the acute post-prandial bioactive effect on bone metabolism and should not be interpreted as a predictor of long-term outcome on bone health. Qualifying an interventional change in bone health by change in bone mineral density may only be detected following a 3-to-6-month intervention and is only of relevance to the specific population under investigation [32].’

  1. Long-Term Impacts on Bone Health: The study assesses short-term bioavailability, which does not provide information on the long-term effects of the supplement on bone health, including bone remodeling and changes in bone mineral density. Prolonged metabolic changes influence osteoporosis; therefore, a brief acknowledgment of this limitation and a call for long-term studies would add significant value to the paper.

Hopefully this has been addressed in the revised limitations section lines 362 through 373.

  1. Complex Mechanisms Underlying Osteoporosis: Osteoporosis is a multifactorial disease involving not just calcium deficiency but also hormonal, genetic, dietary, and lifestyle factors. Discussing these complex mechanisms, particularly the influence of sex hormones and other factors like vitamin D levels, would enrich the context of your findings and underscore the multifaceted nature of bone health.

Response: Like the arguments presented above, expanding the discussion to address adequately factors affecting a multifactorial disease resides outside of the scope of the present study objectives and non-generalisable outcomes.

Detailed Limitations of the Study: Although the manuscript mentions the small sample size as a limitation, a dedicated "Limitations of the Study" section could further elaborate on other constraints, such as the metabolic diversity of participants, lack of control over dietary factors, and limitations of the bioavailability measurement methods used. Highlighting these aspects would clarify the study's boundaries and reinforce the necessity for meticulous and comprehensive future research.

Response: Agreed. We have attended to this in the revised m/s, adding a separate section on the study limitations, lines 661 through 372 and reinforced in the concluding statement, lines 374 through 390

Recommendations for Minor Revisions:

Response: Addressed in the above.

  • Expand the discussion to address the impact of supplementation on overall mineral metabolism, including magnesium, phosphorus, and zinc.
  • Address the potential differences in calcium bioavailability between plant and animal proteins and suggest further research in this area.
  • Discuss the limitations of the study population, particularly the focus on young women, and propose future studies involving peri- and postmenopausal women.
  • Highlight the importance of long-term studies to assess the effects of the supplement on bone remodeling and mineral density.
  • Include a broader discussion on the complex mechanisms contributing to osteoporosis, emphasizing hormonal and other non-calcium-related factors.
  • Develop a more comprehensive "Limitations of the Study" section that details sample size constraints, metabolic variability, and methodological limitations.

Reviewer 2 Report

Comments and Suggestions for Authors

The authors made a great effort to reply to the criticisms raised by the reviewers. However, the methods' description should be carefully improved.

-It's still not crystal clear to me the number of the recruited individuals. Table 1 must be improved including the real number of the recruited men and women in the CON and SUPP groups. Mean values, SD, and the relative ps should be updated too.

- The authors should state in the paper that they did not find sex differences in their results, or they should show putative differences if disclosed.

- In the file I downloaded, Figure 1 is missing the description in the ordinate axis.

-The captions for figures 1 and 2 are staggered. Please check.

-Section 3.4 and Figure 4 do not show the data for CON and SUPP. Why?

Author Response

Response to reviewer # 2 2nd review: Comments and Suggestions for Authors

The authors thank the reviewer for taking the time to re-review this M/S

Comments and Suggestions for Authors

The authors made a great effort to reply to the criticisms raised by the reviewers. However, the methods' description should be carefully improved.

Response: An encouraging viewpoint on the 1st review. But …..

-It's still not crystal clear to me the number of the recruited individuals.

The authors may have located the source of the reviewer’s concern, residing within the CONSORT flow chart. There is an error in this chart. The number block randomized to each of the 2 study arms is stated as n=10, when it should be stated as n=5 on cross-over. This has now been corrected. Apologies for the error. Hopefully this was source of the concern? 

Table 1 must be improved including the real number of the recruited men and women in the CON and SUPP groups. Mean values, SD, and the relative ps should be updated too.

Response: The authors find it difficult to clarify further w.r.t. Table 1 as the number of participants (n=10), 5 men and 5 women is stated in the header to the Table, i.e., Table 1. Participant characteristics (n = 10; 5Ơ, 5Ǫ). These are the ‘real’ numbers. To assist, the number and sex of participants has been added to the text as follows.

Line 92 .. ‘Ten participants were recruited to this study, five men and five women.

Mean and SD values are included in Table 1. It is unclear why ‘relative ps(??) should be updated’??

- The authors should state in the paper that they did not find sex differences in their results, or they should show putative differences if disclosed.

Response: As stated previously ‘The study was a within-subject, single-blind, block randomised, repeated measures, cross-over design comprising two study arms control (CON) and supplement (SUPP), i.e., subjects acting as their own control. The study makes no prior assumption as to differences that may be attributed to confounding factors such as the participant’s sex or lifestyle (e.g. diet, physical activity, etc.).’

To further endorse the study design and subject eligibility (stated in lines 85 to 86 in the M/S) and recruitment the following has been added (Section 2.2, line 99)

‘The study was open to young adult men and women, independent of sex, who satisfied the inclusion criteria.’

Thus, what transpires to be a balanced outcome, i.e., 5 men and 5 women was not by study design. The authors do not consider it wise to retrospectively engage is an analysis of sex-based difference in response as (i) it was not stated as a primary objective and (ii) the number of subjects is too small.

- In the file I downloaded, Figure 1 is missing the description in the ordinate axis. The captions for figures 1 and 2 are staggered. Please check. Response: This appears to be a formatting issue in the positioning of the Figure.

-Section 3.4 and Figure 4 do not show the data for CON and SUPP. Why?

Response: It was reasoned that only SUPP, due to its nutrient content (absent in CON), would ‘bioactive’ with respect to the enteroendocrine response.